# Multi-View Learning to Unravel the Different Levels Underlying Hepatitis B Vaccine Response

**DOI:** 10.3390/vaccines11071236

**Published:** 2023-07-13

**Authors:** Fabio Affaticati, Esther Bartholomeus, Kerry Mullan, Pierre Van Damme, Philippe Beutels, Benson Ogunjimi, Kris Laukens, Pieter Meysman

**Affiliations:** 1Adrem Data Lab, Department of Computer Science, University of Antwerp, 2020 Antwerp, Belgium; 2Antwerp Unit for Data Analysis and Computation in Immunology and Sequencing (AUDACIS), University of Antwerp, 2020 Antwerp, Belgium; 3Centre for Health Economics Research & Modeling Infectious Diseases (CHERMID), Vaccine & Infectious Disease Institute (VAXINFECTIO), University of Antwerp, 2610 Antwerp, Belgium; 4Antwerp Center for Translational Immunology and Virology (ACTIV), Vaccine and Infectious Disease Institute, University of Antwerp (VAXINFECTIO), 2610 Antwerp, Belgium; 5Centre for the Evaluation of Vaccination (CEV), Vaccine and Infectious Disease Institute, University of Antwerp, 2610 Antwerp, Belgium; 6Department of Paediatrics, Antwerp University Hospital, 2650 Edegem, Belgium

**Keywords:** immunoinformatics, integrative algorithms, predictive framework, interpretable framework, systems vaccinology

## Abstract

The immune system acts as an intricate apparatus that is dedicated to mounting a defense and ensures host survival from microbial threats. To engage this faceted immune response and provide protection against infectious diseases, vaccinations are a critical tool to be developed. However, vaccine responses are governed by levels that, when interrogated, separately only explain a fraction of the immune reaction. To address this knowledge gap, we conducted a feasibility study to determine if multi-view modeling could aid in gaining actionable insights on response markers shared across populations, capture the immune system’s diversity, and disentangle confounders. We thus sought to assess this multi-view modeling capacity on the responsiveness to the Hepatitis B virus (HBV) vaccination. Seroconversion to vaccine-induced antibodies against the HBV surface antigen (anti-HBs) in early converters (*n* = 21; <2 months) and late converters (*n* = 9; <6 months) and was defined based on the anti-HBs titers (>10IU/L). The multi-view data encompassed bulk RNA-seq, CD4+ T-cell parameters (including T-cell receptor data), flow cytometry data, and clinical metadata (including age and gender). The modeling included testing single-view and multi-view joint dimensionality reductions. Multi-view joint dimensionality reduction outperformed single-view methods in terms of the area under the curve and balanced accuracy, confirming the increase in predictive power to be gained. The interpretation of these findings showed that age, gender, inflammation-related gene sets, and pre-existing vaccine-specific T-cells could be associated with vaccination responsiveness. This multi-view dimensionality reduction approach complements clinical seroconversion and all single modalities. Importantly, this modeling could identify what features could predict HBV vaccine response. This methodology could be extended to other vaccination trials to identify the key features regulating responsiveness.

## 1. Introduction

The immune system is a complex network of interconnected components, including effector cells, receptors, and mediators that underlie both humoral and cell-mediated immunity and innate and adaptive responses [1]. A single-level analysis of any of these components fails to fully capture the heterogeneity of the immune system and may not uncover meaningful markers. To address common research questions, a “systems immunology” approach is often necessary to leverage the interrelationships between different component levels. This framework aids in detecting shared patterns and characterizing the drivers of immune responses in a comprehensive manner [2,3,4]. Unfortunately, few methods are available within systems immunology that can directly interrogate immunological data across different levels.

In recent years, the field of biomedical research has witnessed exponential growth in data availability, complexity, and heterogeneity stemming from advancements in high-throughput technologies. To reap the benefits hidden within these data, traditional analysis methods alone do not suffice. Machine learning has emerged as a powerful tool for biomedical researchers to automatically learn from and make predictions based on data. By exploiting this technology, researchers can handle multidimensional datasets with data-driven strategies that are tailored to biomedical problems. Through artificial intelligence, the inherent heterogeneity and noise often encountered can be tackled to extract valuable insights that are otherwise difficult to discern, holding great promise for clinical applications.

Multi-view learning is a branch of machine learning techniques that consist of the fusion and integration of multiple datasets. By exploiting the additional intra-dataset information available, models thus built can aim for a better generalization performance [5]. Multi-view learning has received a lot of attention, and several algorithms with encouraging performance have since been developed, with many based on Canonical Correlation Analysis (CCA) [6]. Multi-view learning has been successfully applied to biological data integration. In particular, CCA-based applications have been implemented for oncological, Alzheimer’s studies, and brain imaging tasks [7,8,9,10,11].

Principal Component Analysis (PCA) [12,13] is a more well-known dimensionality reduction technique that computes a linear transformation of original features to produce a new set of orthogonal features of lower dimensionality that can capture maximal variability in the data. These derived variables are uncorrelated and placed in descending order of their explained variance. CCA belongs to the same category of statistical techniques as PCA and could be qualified as Joint Dimensionality Reduction (JDR). The main difference between these two methods lies in the input and the optimized statistics. While PCA operates on a single multivariate dataset at a time, CCA works on two datasets. CCA has since been modified to function on more than two datasets at once and exploit their interdependencies by maximizing the sum of pairwise correlations (SUMCOR) [14]. The dimensions of these different views can be simultaneously reduced while maximizing their correlation across modalities instead of variance, irrespective of the data types.

The modeling of immunological datasets is, unfortunately, restricted to the common multi-omics data integration approach, which does not make use of CCA. This multi-omics method consists of a fine-tuned design that is aimed toward a specific set of data types, with Multi-Omics Factor Analysis (MOFA) as a prime example [15]. MOFA allows the direct use of relational information to integrate the levels; however, its specialization locks their applicability to datasets in their field and hinders their use in immunology where other levels are more commonly measured, such as the immune cellular composition or T cell and B cell receptor (TCR/BCR) repertoires.

Multi-view analysis is potentially beneficial to the vaccinology field. As highlighted in the literature, many factors influence the efficacy of vaccinations. These factors comprise identifying immune cells that are critical for a robust immune response, including which memory cell population needs developing (e.g., B cell vs. T cells) and understanding how patient factors (e.g., gender, age, ethnicity) influence these findings. For instance, age significantly modifies the landscape of the immune system. These aging changes are well documented in age-related pathologies such as Alzheimer’s, arthritis, hypertension, and atherosclerosis, which are linked to chronic inflammation and interferon activation [16,17,18]. Additionally, pre-existing cross-reactive T-cells, such as in the CD4+ memory compartment, are required for a robust vaccine response to the Hepatitis B virus (HBV) [19]. Therefore, multi-level factors need to be considered when determining who would respond best to a vaccination. However, the currently applied approaches only analyze one layer at a time, and hence, might lead to potential confounders when interpreting the data.

Among the unsolved questions in vaccinology is what drives and predicts a good immune response to a vaccine. Prior studies have shown that a vaccine response can, in part, be predicted using mRNA expression baseline levels from blood (cells) [20,21,22]. Thus, the response to a vaccine seems to be partially determined by the state of the immune system before the vaccine is administered. Predictive models based on a single data type, such as peripheral blood mononuclear cells (PBMCs), bulk transcription RNA expression, or flow cytometry (protein expression), have been previously built. However, as indicated before, these different measurement levels are not independent, with ~40% of mRNA levels correlated to protein expression [23]. Therefore, it is not known what is causative and what is a related confounder. A multi-level approach may aid in deconvoluting these potentially confounding factors.

In this paper, we evaluate different multi-view approaches on multi-view data, including bulk blood RNA sequencing (RNA-seq) gene expression, PBMC flow cytometry cell counts, memory CD4+ T-cell receptor repertoire, and demographic metadata with reference to the HBV vaccine response. We aimed to study whether this methodology could be helpful in determining the predictability of the baseline state for the vaccine response. As a proof of concept of our methodology, here we highlight multi-view learning and identify the biological markers that correlate with the response profiles developed after de novo HBV vaccination. These markers, deriving from different data types, include clinical factors such as age and gender, as well as inflammatory gene sets and vaccine-specific T-cells that predate the vaccination challenge.

## 2. Materials and Methods

### 2.1. Study Cohort

The study participants had been previously recruited to assess HBV vaccination response [19,24]. The cohort consisted of 34 HBV-naïve individuals who were neither vaccinated nor infected (according to serology) against/by HBV. The original publications aimed to identify a response to a three-dose regimen of the Engerix-B vaccine, with second and booster doses, respectively, administered 30 days and a year after the first dose. The vaccine response was measured based on antibodies against HBV surface antigen (anti-HBs) titers and was captured by ELISA assays at days 0, 60, 180, and 365 (summarized in Table 1). To assess seroconversion and thus presumed protection, a threshold of 10 IU/L was commonly considered [19]. For each patient, if this value was reached before two months, early conversion was determined. If this value was induced prior to 6 months, late conversion was registered for the individual. The patient was otherwise considered a non-converter.

For the non-converter class, it was impossible to determine if seroconversion was eventually reached. Moreover, TCR data were available for only three non-converters. Therefore, this class was even more limited in size, further hindering the use of upsampling techniques to balance the task. Non-converters were thus excluded from further analysis.

The metadata of each participant was also considered (Appendix A). All clinical features were collected before administering the first dose. The features included age, gender, body temperature, and maximum and minimum blood pressure (Min_BP and Max_BP). Age distribution differed across all 3 classes (Figure 1), with late converters tending to be older than the other groups (Appendix A).

Other clinical factors that were available included access to the absolute numbers of white blood cells (with differentiation in monocytes, lymphocytes, and granulocytes), red blood cells (with separate attributes for hemoglobins and hematocrit), and platelets, which were determined with a hematology analyzer [19,24] (Appendix A). Lastly, flow cytometry identified the percentages of CD4+ T cell relevant information (Appendix A), as this population was critical to developing a robust HBV antibody response. These CD4+ T cell parameters included the normalized ratio of vaccine-specific T-cells (HepBTCR), the total number of TCRs sequenced in CD4+ memory (B0), the frequency of bystander TCRs (PPnrB0), and the frequency of vaccine-specific TCRs (PSB0).

### 2.2. RNA-Seq Data Quality Control Steps

The first study [19] detailed the blood extraction and bulk RNA-sequencing of PBMCs at day 0 and after vaccination at days three and seven. As stated, the pre-vaccination time point was considered. Of the 23,812 transcripts, gender-specific and hemoglobin genes, as well as removed transcripts with fewer than 100 counts across the samples, were excluded, which left 13,012 expressed transcripts for analysis. The counts were normalized using DEseq2′s median ratios to correct for the sequencing depth and RNA composition [25] and to allow reliable between-sample comparisons. Differential gene expression analysis was performed with DESeq2 negative binomial models. To improve the interpretability, the gene counts were aggregated based on the published BloodGenModule3 features to reduce the high dimensionality effects that this data view could carry in the analysis [26]. BloodGenModule3 was chosen, as it contained a hierarchical, fixed repertoire of 382 functionally annotated gene sets (blood transcriptional modules [BTMs]) [27]. Aggregation BTMs by their means were used.

### 2.3. Data Sampling

The class distribution was relatively unbalanced. Synthetic Minority Oversampling Technique (SMOTE) [28] was used to up-sample the minority class (late-conversion) in the training data set at each iteration of the cross-validations used to estimate the performance. It has been shown that SMOTE can be beneficial to high-dimensional datasets when applied after feature reduction, such as in this case. New data points were generated by computing the average measures of the five nearest neighbors of the same class in the training data set.

### 2.4. Integration Methods

The 30 early and late converter samples were used to evaluate several integration approaches for their applicability to the four processed views (Appendix A). The workflow is detailed in Figure 2. Each data view was also initially input into the Logistic Regression (LR), the multivariable model commonly used for biological sciences studies, to establish a baseline and examine the separate contributions. LR coefficients were readily interpreted, as the regression model required few assumptions to determine the relationships between the dependent and independent variables [29].

The simplest and most common method to implement was an early fusion in which the views were concatenated at a feature level, and a model was trained on all the variables. However, the advantageous ease of application was counterbalanced by the likely skewed fusion caused by high dimensional sets, such as our gene expression matrix [8,30]. Mirroring this first technique, a late fusion through voting was also applied [31]. The concatenation, in this case, happened at the decision level by merging independent predictions through weighted averaging. A classification model was built to exploit specific characteristics of the target view. These predictions were then weighted and combined in the final results.

Feature engineering was explored next. Variants of PCA were deemed to be most appropriate for our case. A sparse PCA variant [32,33] that introduced a Least Absolute Shrinkage and Selection Operator penalization [34] was applied on the concatenated features, and its output fed was as training to the models (named “PCA_singleview”). A PCA was also applied to each separate view to discover equivalent dimensions of the view-specific components. In the case of the gene expression set, the sparse variant was used. The embeddings obtained were then concatenated along the component dimension before training (named “PCA_multiview”). As an even more compact alternative, each view was whitened with PCA; then, latent features were concatenated, and a PCA was performed on this matrix once more (Group PCA) [35]).

An adapted multi-view version of CCA, Multi-viewCCA (MCCA) [14,35,36], in its regularized form, was implemented. The regularization parameter was introduced to reduce the overfitting caused by differences in dimensionality between the datasets. Regularized MCCA aimed to discover a common signal across sources while also reducing view-specific noise [37]. Sparsity was introduced to denoise the gene expression in particular [38]. The number of features in this case greatly exceeded the number of observations, and CCA could not be applied otherwise. Shared components could then potentially show meaningful discriminant features. Similar to PCA, each component could be individually studied to reveal the factors that most contributed to it. This aspect was fundamental to drawing valuable insights from the task at hand. The decision to reduce the summed dimensionality (399 features) of the multi-view data to a few dimensions allowed us to obtain substantially more interpretable results.

Some of CCA’s assumptions were considered during our analysis. Even though it was not strictly necessary for the machine learning algorithms used here, we applied standardization on every dataset to respect CCA’s assumption of multivariate normal distribution and the homogeneity of variance. The relationship between the canonical variates calculated and the original features was assumed to be linear unless a kernel variant was used. Similar to multivariate regression, and many other statistical algorithms, CCA gained inference power and reliability from a high sample size. The application of SMOTE thus contributed to the generation of more data points. These assumptions were also applicable to every other PCA-based transformation we implemented.

## 3. Results

### 3.1. Anti-HBs Titers Separation

The anti-HBs conversion status was assessed based on the three-time points after vaccination (Figure 3A). A PCA transformation of the antibody titer data revealed that principal component (PC) 1 (93% of the variation) segregated the data based on when the individuals responded to the vaccination. As expected, a clear separation occurred between the three clinically relevant groups: early converters, late converters, and non-converters (Figure 3B).

### 3.2. Independent Data Levels Show Correlated Features

The collinearity across views was an expected characteristic within the multi-view HBV vaccine data set, as each molecular or clinical level influenced the others. This step aimed to uncover dependencies within the metadata irrespective of the vaccine response. As expected, several variables were strongly correlated (Figure 4). For example, the day 0 lymphocytes (LYM0) and granulocytes (GRA0) counts were strongly anti-correlated (R = −0.98). As per the literature [38], the expected negative correlation of the female gender (Gender_F) and red blood cell count (RBC0) (R = −0.63) was identified. TCR-derived parameters were also assessed. The day 0 total unique TCR sequences per individual (B0) was also correlated with both the fraction of HBs-Ag-specific (PSB0, R = 0.88) and non-specific(PPnrB0, R = 0.61). The normalized ratio of vaccine-specific TCRs (HepBTCRs) was poorly correlated to the initial TCR counts (B0; R = 0.16).

### 3.3. Individual Levels Contain Elements That Correlate with Response

Before implementing the multi-view integration, we interrogated the data to identify the key features. Foreseeably, age was higher for late converters (*p*-value = 0.019, Appendix A). Age was also anti-correlated to the anti-HBs titers presented at each time point, especially after the first two doses (Appendix A). Within the flow cytometry modality, lymphocyte counts seemed to possess a positive association with early response (Appendix A). On the other hand, white blood cells and, in particular, granulocytes were associated with a delayed response (Appendix A). However, possibly due to outliers, no strong statistical significance emerged from these comparisons. Conversely, the testing of the ratio HepBTCR revealed a significant association with an early HBV vaccine response, which was traceable to the higher HBV specificity of TCRs (*p*-value = 0.003, Appendix A). Just one gene was identified as differentially expressed prior to vaccination when comparing the early and late converters: HLA-DRB5 (Log_2_FoldChange = −4.96, adj. *p*-value = 2.50 × 10^−4^ [False discovery rate = 0.05]). HLA-DRB5 was down-regulated in the late converters and encoded a part of the MHC class II complex in relation to antigen presentation and CD4+ T cells.

### 3.4. Projection of Data Views Provides Insights into Response Classes

The different data levels were integrated and projected using two distinct approaches (Figure 5). The first was a PCA, where PCs were defined by those that captured the most variance across the entire concatenated dataset. The first two PCs could be evaluated through the visualization of the samples in the lower dimensionality latent space (Figure 5A). The second was MCCA, which aimed to detect a common latent structure across multiple datasets by following the SUMCOR definition. The two most informative canonical variates after same-time MCCA integration could then be inspected (Figure 5B). The MCCA approach experienced more prominent separation in the data.

As MCCA had a better separation of the converter status, it could further be explored by considering the loadings underlying the new dimensions, which represented the contribution of each base feature. Appendix A shows the stacked coefficients for each of the two first canonical variates of MCCA, grouped by function. These effects could be attenuated if GroupPCA was considered (Appendix A).

There are certain pre-existing features that are associated with vaccine response [19], including specific gene expression patterns (e.g., cytokine expression), immune cell overrepresentation, and patient factors (e.g., age, gender, comorbidities). Upon inspection of the first component of the MCCA projection age, the oxidative stress, inflammation, and interferon modules all had negative contributions (Appendix A). Lymphocytic BTMs presented a negative sign to their coefficients as well. Furthermore, erythroid cell modules and blood cell-related counts were relevant for both dimensions and, in particular, for the second component (Appendix A). As could be expected based on the prior pairwise correlations, higher red blood cell counts were also associated with the male gender parameter [39]. The signal given by monocytes was weaker but more consistent for the second component, with the monocyte count and monocyte-related modules showing a positive contribution. Finally, CD4 lymphocytes were present for the contribution of the two components both in terms of the modules and counts that came from the flow cytometry assays. This compound observation reinforced the importance of lymphocytes for projection.

Moving to GroupPCA, we could identify, for the first PC (Appendix A), several age-related markers. In the present study, inflammation, interferon, oxidative stress, and phosphorylation signatures appeared positively tied to aging. Other positive interactions included higher min and max blood pressure and a higher cell death-driven gene expression as well as lower RBC0, hematocrit (HCT0), and hemoglobin protein counts (HGB0). Finally, the erythroid cells’ BTM expression was noticeably relevant for this component. Platelet counts and BTMs appeared negatively related to age.

For the second PC (Appendix A), at lower ages, we also observed lower blood pressure, reduced cell death, oxidative stress, and phosphorylation. Diminished T cell and lymphocyte-related features seemed to signal a less active adaptive immune system. On the other hand, heightened white blood cell counts, neutrophils, and neutrophil activation modules, including granulocyte counts, together with the presence of inflammation modules, still showed innate immunity characteristics for this dimension.

In comparison to MCCA, GroupPCA revealed distinct modules (Appendix A). The top homogenous contributions included the interferon and inflammation modules. This was distinct from MCCA (Appendix A), where these features were not represented as having coherent contributions. GroupPCA presented with a stronger T cell signal. Both methods assigned rather consistent importance to gene transcription, protein synthesis, and protein modification modules for both of their dimensions.

Additionally, to complement the previous analysis and to disentangle the loadings that presented conflicting contributions, features significantly correlated to one of these latent components (*p*-value < 0.05) could be overlaid on top of the MCCA projected cohort. This visualization allowed us to identify a cohesive inflammatory signature. A region of the latent space was thus univocally characterized by elevated temperature, age, granulocyte counts, inflammation, and neutrophil-related gene sets (Figure 6B). As a result of the inferior class separation, only a more ambiguous sub-setting of the cohort could be derived from visualizing correlations of the same features on the standard PCA (Figure 6A).

### 3.5. Multi-View Integration Allows Superior Classification Performance

Thus far, the integration of this data set has been explored in an unsupervised fashion, i.e., without prior knowledge of the converter classes. However, as these were known, the various multi-view integration methods could be evaluated for their ability to capture and predict this ground truth. This was conducted by feeding the integration output into a supervised machine learning model with a Leave-One-Out-Cross-Validation (LOOCV) approach. Integration approaches that successfully captured latent patterns within the data set that were related to vaccine response should be able to obtain high performance.

The LOOCV was repeated 20 times with random initializations to identify a more reliable performance spread (Figure 7). The classification task was originally skewed with a majority class of twenty-one early converters versus a minority class of nine late converters. The evaluation of each data layer independently with an LR (Appendix A) revealed that not all layers held the same predictive capacity. Overall, the TCR layer had the strongest predictive performance on validation data, with both balanced accuracy and area under the receiver operating characteristic curve (AUC) measures above 0.7 (**0.708 ± 0.067** accuracy; **0.731 ± 0.040** AUC). Conversely, the outputs produced using cell counts or metadata were slightly better than a random classifier, corresponding to a prediction accuracy or AUC that was marginally above the threshold of 0.5. The model using gene expression data consistently assigned a prediction with confidence (>0.6 decision value; Appendix A). This resulted in a higher AUC (**0.611 ± 0.009**) but a lower balanced accuracy (**0.440 ± 0.018**).

To boost the performance of these models, joint dimensionality reduction algorithms were applied. As expected, it improved the LR fitting for both GroupPCA (**0.750 ± 0.034** AUC; **0.655 ± 0.051** accuracy) and MCCA (**0.731 ± 0.042** AUC; **0.627 ± 0.047** accuracy) (Figure 8A). GroupPCA surpassed both the single-level baselines and the early fusion while also adding the interpretability highlighted in the previous section. However, using regular PCA methods was not as effective or even counterproductive, causing, in some instances, a drop in performance compared to the best single-view predictors (Figure 8A).

The data integration techniques produced consistent results for the more complex model, Random Forest (RF) (Figure 8B). Once again and alongside GroupPCA, MCCA presented itself as the best integration method. JDR implementations fell in the range of 0.64–0.8 for both AUC (**0.743 ± 0.053** GroupPCA; **0.770 ± 0.087** MCCA) and accuracy (**0.646 ± 0.079** GroupPCA; **0.640 ± 0.082** MCCA), performing comparatively well, albeit presenting a larger spread. In this setting, the PCA single view was the only method to present a poor performance. After such a strong dimensionality reduction, this result was expected. The models were effectively trained on very few features, and a simpler method should be sufficient in this circumstance.

In addition, RF and LR could be easily unraveled to further allow for the interpretability of underlying predictions. The coefficients of the LOOCV LR were trained on single data views (Figure 9). Old age was the leading factor for a late response (negative class) in the unimodal analysis (Figure 9C), which was also apparent in our previous modeling.

Here, feature importance within the dataset was showcased, which included granulocyte counts, neutrophils modules, and other immune-related BTMs linked with late response predictions. These features were distinct from the earlier converters, which included the female gender, temperature, pre-existing vaccine-specific T-cells (Figure 9B), platelets count and BTMs, the monocytes count, as well as erythroid cells associated counts and modules. The coefficients for platelet counts, erythroid cells and platelet BTMs, presented the highest median for the single-view flow cytometry, and RNA-seq models, respectively (Figure 9A,D).

## 4. Discussion

Here, we showcased the utility of using multi-view analysis to aid in the interoperability of a longitudinal dataset following the response to HBV vaccination. This analysis involved incorporating anti-HBs titer levels, RNA-sequencing, the CD4+ T-cell receptor, flow cytometry, and patient-defined features (e.g., age, gender). This multi-view analysis implementation aided in finding interpretable patterns in a vaccination context, which had not been previously interrogated together. This study based itself on the notion that prior studies have also highlighted relations between different levels (e.g., mRNA expression, CD4+ T-cells, age, and BMI) and the eventual HBV vaccine response, as measured by anti-HBs titers [19,24,40,41]. To point out the most meaningful approach and improvement in the predictive power of multi-view datasets, we used several integration methods. Our approach identified the key baseline features associated with earlier and later converters after HBV vaccination. This modeling was able to make informed interpretations of the data and account for confounding factors. For instance, our modeling was able to highlight that older age and B-cells were more associated with late converters.

Age is undoubtedly a major factor in immune response, especially with “inflammaging” and “immunosenescence” being well-documented phenomena [16,17,42]. The impact of age is so broad that, for instance, even though monocyte numbers have not been shown to differ in older populations when compared to younger subjects, their phenotype, and functionality are considerably different, with a higher degree of senescent qualities [43]. Inflammation, interferon, oxidative stress, and phosphorylation dysfunctions in the elderly were also widely confirmed by the literature [44]. Even though the age distributions of early and late converters were not substantially dissimilar, the data indicated a tendency that supported the argument of a higher risk of malfunctioning in the immune system of the elderly. Furthermore, the negative association between seroconversion and B-cell-related gene expression might seem counter-intuitive, given their role as antibody producers. However, this finding could be reconducted to the overall consideration that an activated immune system at the baseline is, at least partially, responsible for a poor response to vaccination. In our study, these signatures appeared along with higher min and max blood pressure and a higher cell death-driven gene expression. Lower RBC0, HCT0, and HGB0, in conjunction with the positive contribution of age, allowed us to infer that these patients had aging anemia and reduced hemoglobin [45].

Granulocytes and lymphocytes appeared mutually exclusive, showing a very strong anti-correlation: a behavior that was also detected in the literature [46]. According to [46,47,48], a high neutrophil (granulocytes)/lymphocyte ratio was tied to the inflammatory status of the patient and to impaired cell-mediated immunity. Conversely, a lower ratio could be associated with better outcomes for several diseases, including HBV. Likewise, higher temperatures could have a limiting effect on viral replication and play an enhancing role in many immune-related functions [49]. Our model contextually showed both patterns to be important for conversion classification.

Gender presented itself as another important factor for the interpretation of results. The male gender, relevant in this study, has previously been shown to influence the presence of pro-inflammatory markers [50]. Additionally, females have been documented to have a better response to the HBV booster vaccination with greater anti-HB titers [51]. The models presented here showed that the male gender was associated with inflammation-related features and was predictive of an early response when controlling for other variables during regression.

For early response predictions and in accordance with the results for other pathogens [52,53], pre-existing vaccine-specific TCRs contributed positively. Our analysis, through T-cell gene expression and CD4, count contributions, highlighted the potential existence of the pre-vaccination priming of the T-cell compartment, which could be conducive to an early vaccination response, and requires follow-up interrogations on which CD4+ T cells can contribute to vaccine response. Using this modeling approach, we also identified that erythroid cells and platelets counts, as well as their respective BTMs, were also impactful for an early seroconversion. Intriguingly, platelet counts and BTMs were negatively correlated with age and had opposite contributions on single-view model predictions, as a behavior confirmed by the literature [54]. These data could be interpreted as a reduced number of immune cells simultaneously circulating in the blood, leading to an overall less activated system.

Among the dimensionality reduction techniques, GroupPCA was on par with MCCA, surpassing every other method and all four of the single-level models in terms of AUC. However, the fine-tuning of the regularization or the addition of learned weighting parameters to MCCA’s algorithms can boost its performance. Furthermore, MCCA seemed to find a parsimonious representation of heterogeneity in the sample (significant latent space). The separation it produced was more prominent and, we could assume, more trustworthy than single level labeling.

Our modeling achieved a reasonable level of accuracy and AUC; however, this did not reach a median AUC > 0.8, which highlighted that improvements could be made with further refinements. For instance, a parameter of “importance” could be learned through self-attention mechanisms for each view to weigh the contributions differently when applying JDR instead of applying an imposed regularization parameter. Starting from the assumption that the similarity between the samples was different in different views, [55] proposed a smooth representation by using a self-weighting method that enhanced similarity grouping effects. An explanation for the reduced AUC could be attributed to an incorrect converter classification, as this was based solely on the antibody titers. However, our data identified screening for memory CD4+ T cells could be identified if someone responded to the vaccination. Additionally, due to the limited understanding of all possible gene–gene networks (gene modules), this study’s modeling could not map the contribution of genes with an unknown biological function. Therefore, alternative ways to identify the most relevant gene sets are needed. Future versions of the method could have the capacity to include more gene modules, other than [27], and allow the use of first dimension PCA instead of the BTM mean to allow the capturing of dataset-specific variability.

To solve the linearity constraint of these combinations, Kernel MCCA [35] could be used to first project the data in a higher dimensional space via the kernel trick and then apply MCCA to a new feature space. Alternatively, similarity network fusion (SNF) [56] emerged as a specific data integration technique to learn underlying global and local structures when dealing with patient data [57,58]. Producing embeddings with encoders could help us handle missing values and missing views, and not only that but help us even reconstruct these missing aspects. In this field where data are scarce and missing data can skew these findings. At the training time, all available views could be used to learn to embed. During testing or generalization, embedding can help reconstruct the missing views’ information since it is learned and incorporated in the embedding. The successful application of Deep CCA and Deep Canonically Correlated AutoEncoders [59] proved the feasibility. Although these implementations are promising, deep learning models had to be excluded at the time of this study due to the limited sample size.

For the purpose of the analysis, a few drawbacks have to be considered. This includes the non-cohesive nature of the contributions that came from the BTMs. Firstly, the noise of the gene expression data hindered the full readability of their contributions. The high dimensionality aspect of RNA-seq data made the pattern harder to interpret. However, noisy results were expected when dealing with such noisy data. This problem is not unique to this study but is widely understood to be an unsolved challenge. Secondly, an effect akin to Simpson’s paradox [60,61] presented itself when inspecting some of MCCA’s loadings. This known effect, caused by multicollinearity in regression analysis, produced a change in the size or sign of the contributions of two positively correlated variables when considered in association with other variables. The effect warns against the easy interpretation of these results and suggests the need for causal analysis. For example, HCT0, RBC0, and HGB0 were strongly positively correlated, and we expected them to appear with same-sign coefficients in our analysis. Yet, the sign of the relationship was reversed. CD4 lymphocytes were present in the contribution for the two components both in terms of the modules and counts coming from the flow cytometry assays; however, this aforementioned effect limited their unambiguous interpretation. GroupPCA seemingly introduced less of this inversion paradox. On the other hand, it was possible to address this issue by visualizing the correlations between the input features and the projected dimensions. The true relationships between variables could hence be revealed.

Here, our multi-view analysis identified a dynamic and multi-faceted HBV immune response that was linked to many factors for early and late responders. When designing this type of experiment, an important aspect to consider is if the response is a primary T cell vs. B cell response, as some vaccines show a primarily T-cell-based response [62,63]. Furthermore, when designing this type of experiment, one needs to consider the most approximate time points, which may not be applicable to the different kinetics (e.g., mRNA, live attenuated, Toxoid, viral vector, etc.) [64]. The types of primary responder cells can change which time points to collect. Currently, most attempts to predict vaccine immune responses are based on the sole antibody titers. This measurement tends to disregard the heterogeneity of the adaptive immune response and disregard T cell contribution. With an unsupervised approach, JDR has been shown to help uncover a space in which the separation between classes of interest can be more pronounced. However, these findings are currently restricted to the HBV vaccine, and testing for other vaccines is needed. We effectively dealt with an unbalanced phase 1 trial-sized population for this analysis. The feature engineering and data integration steps had a decisive effect both in terms of performance and were also paramount in gaining actionable insights. Ideally, a bigger sample should be used for the validation setting.

## 5. Conclusions

Our multi-view modeling reflected many aspects of what is known about natural HBV infections when giving a holistic and systemic perspective of the HBV vaccination response data. Thereby, this methodology could recapitulate the known findings present in the literature, mirroring HBV response signatures and uncovering novel markers. The results presented throughout this paper suggest that the methodology was valid in identifying biologically important features and how multi-level interrogations could aid in finding the most critical variables. This study’s approach identified that early converters have more in common with those that respond well to natural infection, while late responders are associated with older individuals and an inflammatory phenotype. In particular, an overactivation at baseline of the innate component was detrimental to seroconversion, while an early existing vaccine-specific population in the T cell compartment fostered an earlier response. To confirm the utility of this modeling, we recommend testing the methodology on other HBV vaccination cohorts or in other immunological settings. We envision that this type of framework could become useful for guiding rational vaccine antigen discovery as well as for the use of targeted adjuvants, and we hope it can pave the way for advancements in immunological understanding.

## Figures and Tables

**Figure 1 vaccines-11-01236-f001:**
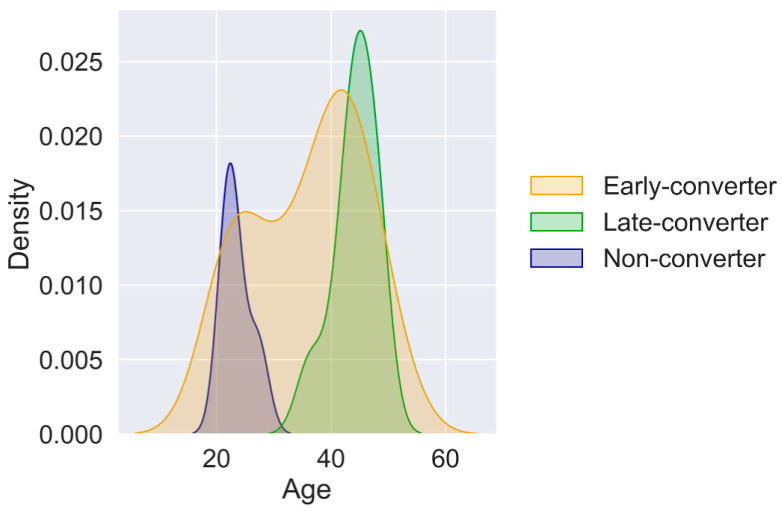
**Age distributions reveal differences between classes**. Average (range) age, in years, per seroconversion class: early converters 35.9 (21.3–50.2), late converters 44.19 (36.3–48.5), non-converters 23.53 (21.6–27.2).

**Figure 2 vaccines-11-01236-f002:**
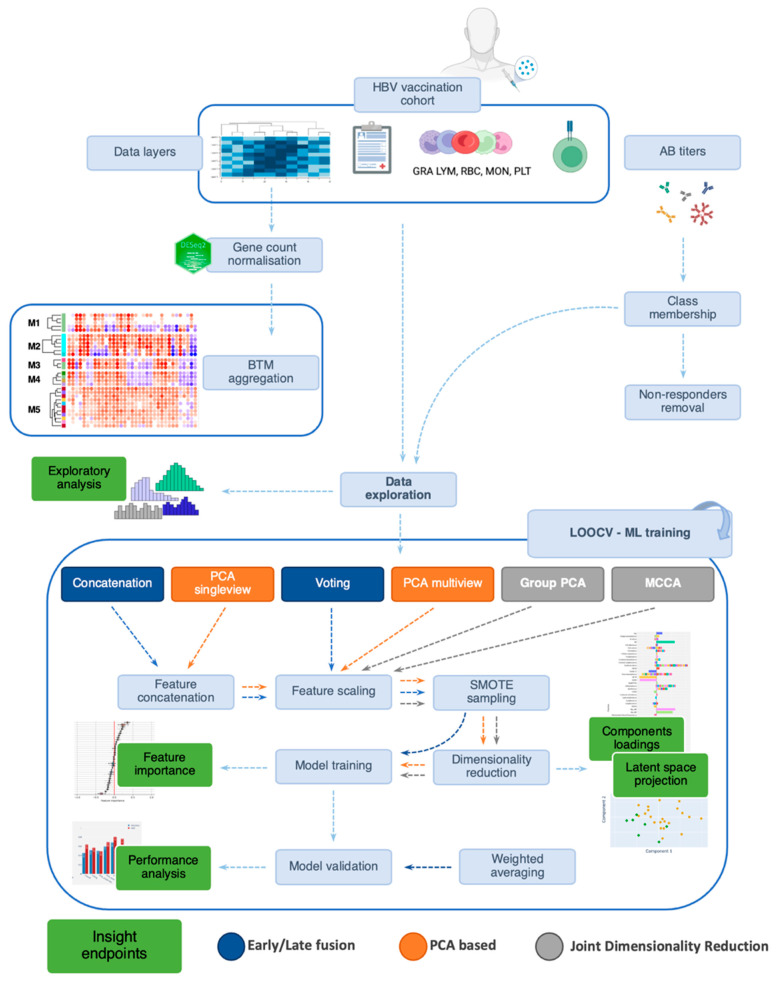
**Research methodology workflow.** The color coding represents the integration philosophies (blue, orange and grey) and, in green, highlights the interpretability endpoints. Antibody (AB); Blood Transcription Module (BTM); Hepatitis B Virus (HBV); Leave-One-Out Cross-Validation (LOOCV); Machine Learning (ML); Multi-view Canonical Correlation Analysis (MCCA); Principal Component Analysis (PCA); Synthetic Minority Oversampling Technique (SMOTE). Figure created using BioRender (BioRender.com accessed on 1 February 2023).

**Figure 3 vaccines-11-01236-f003:**
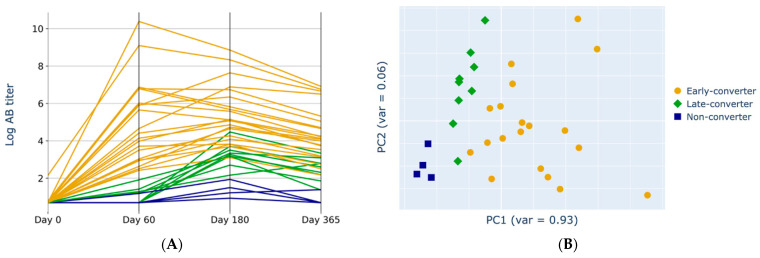
**The Principal Component analysis of anti-HBs titers correlated with responsiveness.** (**A**) Antibody titers at different sampling time points on a logarithmic scale, divided across the three response classes. Lines colored by converter type (early converter = orange; later converter = green; non-converter = blue). (**B**) Principal component (PC) 1 and PC2 of the antibody titer data. Colored by converter type (early converter = orange circle; later converter = green diamond; non-converter = blue square).

**Figure 4 vaccines-11-01236-f004:**
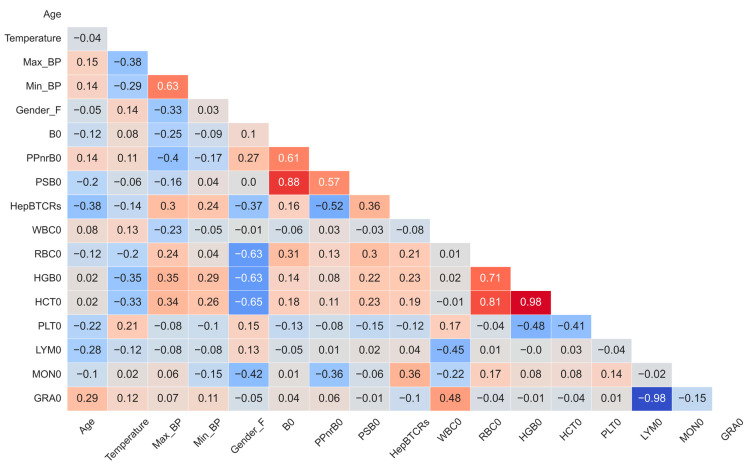
**Correlation matrix shows overlap of cohort features.** Data represent the correlation coefficient across each non-genetic feature (see Appendix A for cohort details).

**Figure 5 vaccines-11-01236-f005:**
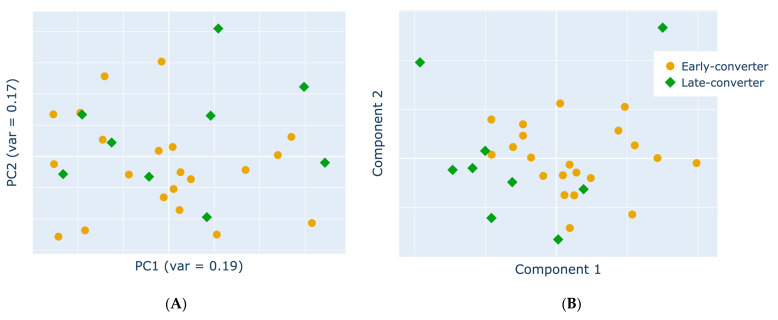
**MCCA latent space projection provides superior class separation.** Two-dimensional data projection with (**A**) PCA and (**B**) MCCA of the integrated multi-view vaccine dataset. Each scatter plot showcases the first two dimensions. Data represent the early converters (orange circle) and late converters (green diamond) after applying each model.

**Figure 6 vaccines-11-01236-f006:**
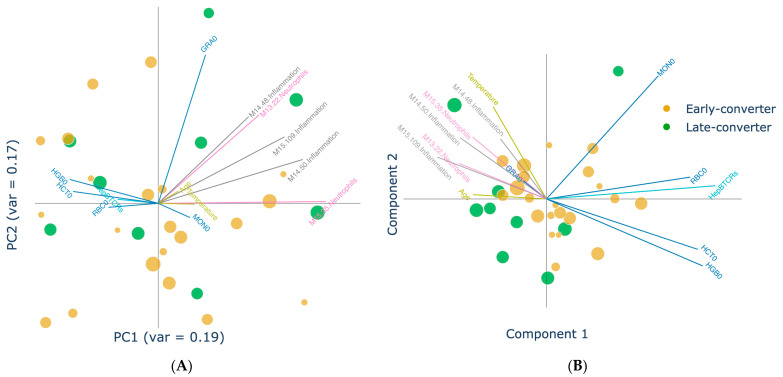
**MCCA latent space projection showing inflammatory markers and characterizing part of the cohort.** Two-dimensional data projection with (**A**) PCA and (**B**) MCCA after the application of a feature correlation overlay. Features significantly correlated to the MCCA-projected space revealed a distinct inflammation-associated region. Marker sizes are proportional to the age of the individual. Feature vector colors identify separate views and module functions. Vector coordinates were calculated using the Person’s correlation for the respective latent component. Patients are colored by converter type (early converter = orange; later converter = green). Granulocytes count (day 0) (GRA0); hematocrit (day 0) (HCT0); normalized ratio of vaccine-specific TCRs (day 0) (HepBTCRs); hemoglobin protein count (day 0) (HGB0); monocytes count (day 0) (MON0); red blood cells count (day 0) (RBC0).

**Figure 7 vaccines-11-01236-f007:**
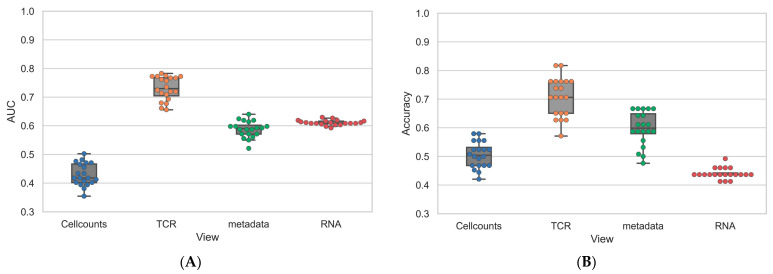
**Single view performance comparison showing the superior predictive power of the TCR-seq layer.** Methods performance comparison for the Logistic Regression classifiers trained on single modality data in terms of (**A**) AUC and (**B**) Accuracy. The colored dots indicate the performance of each of the 20 LOOCV runs. Area under the receiver operating characteristic curve (AUC); T-Cell Receptor sequencing data (TCR); T-Cell Receptor Sequencing (TCR-seq).

**Figure 8 vaccines-11-01236-f008:**
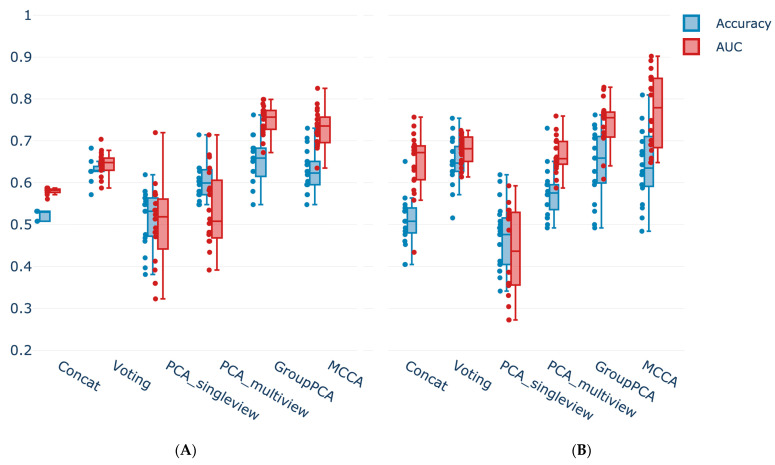
**Classifiers performance comparison showing the positive impact of joint data integration on predictions.** Methods performance comparison for the (**A**) Logistic Regression (LR) and (**B**) Random Forest classifiers in terms of AUC and accuracy. The colored dots indicate the performance of each of the 20 LOOCV runs.

**Figure 9 vaccines-11-01236-f009:**
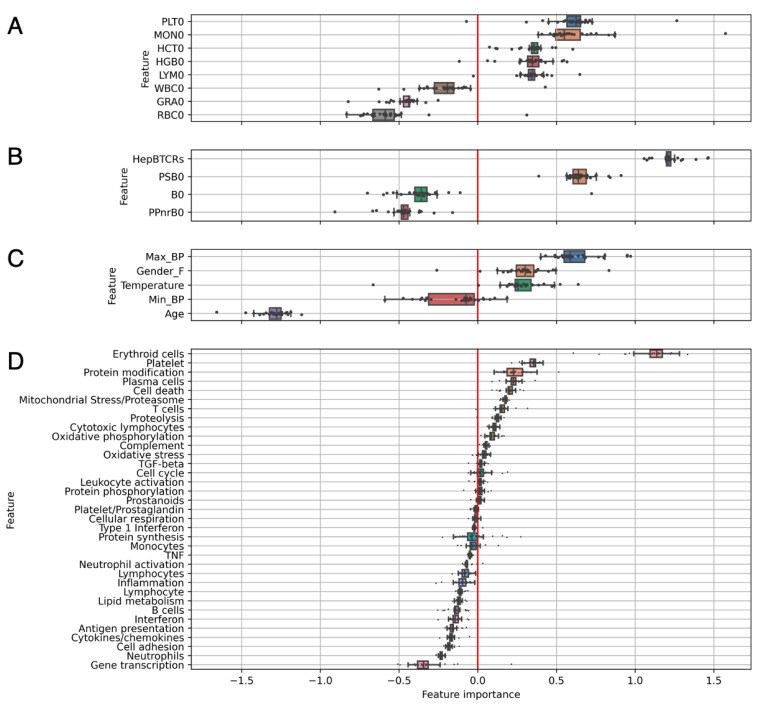
**Unimodal feature importance analysis bolstering insights into biological markers tied to vaccine responsiveness.** Importance of leave-one-out cross validated LR coefficients for the unimodal models: (**A**) Cell counts, (**B**) CD4+ T-cells, (**C**) Metadata, (**D**) BTMs. Positive coefficients favored early response predictions while negative coefficients favored late-response predictions. The vertical red line indicates the zero-importance threshold. BTMs with unknown functionality have been excluded from the visualization. Individual dots indicate feature importance during each of the 20 LOOCV runs.

**Table 1 vaccines-11-01236-t001:** Cut-off for converters and non-converters to hepatitis-B vaccinations.

Outcome	AB Titer	Time	# in Group
non-converters	<10 IU/L	-	4 *
early converters	>10 IU/L	60 days	21
late-converters	>10 IU/L	180 days	9

* Excluded from further analysis.

## Data Availability

The datasets analyzed during the current study and the code used to generate the results are available on GitHub at https://github.com/fabio-affaticati/MultiModHepB (accessed on 17 April 2023).

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
