# Peer review of "Multi-View Learning to Unravel the Different Levels Underlying Hepatitis B Vaccine Response"

_vaccines, 2023, doi:10.3390/vaccines11071236_

Round 1

Reviewer 1 Report

I have found another study in the literature already published by the same authors. Therefore, the manuscript should be rejected. 

Author Response

Point 1: I have found another study in the literature already published by the same authors. Therefore, the manuscript should be rejected.

Response 1: The research has been publicly shared on the preprint server for biology, bioRxiv, but the authors stand by the fact that it has not been previously published in a scientific journal. The manuscript, in its preprint version, can be found at the following link: https://www.biorxiv.org/content/10.1101/2023.02.23.529670v1

Reviewer 2 Report

The authors have tested the ability of several machine learning algorithms to identify factors influencing vaccine efficacy, using the results of a hepatitis B vaccine trial for their inquiry.  They found that multi-view canonical correlation analysis (MCCA) and Group Principal Component Analysis (PCA) were equally superior in their ability to identify pertinent factors.  No real insights into factors influencing vaccination response appear to have resulted from this study.  The utility of the study seems to reside in its being an early attempt to apply machine learning approaches to vaccinology questions that may be useful to refining ML methods in the field.  The authors should add a table of abbreviations.  A better explanation of ML approaches to biomedical problems would also be helpful.  Although it may have been addressed in the original reports of this trial, it would be helpful to describe how HBs titers = 100 relate to HBV protection.  It is surprising that B cells are strongly negatively related to HBV protection (Fig. 9).  A discussion of this would be interesting.  Is it related to methods of assessing B-cell function?

Author Response

The authors have tested the ability of several machine learning algorithms to identify factors influencing vaccine efficacy, using the results of a hepatitis B vaccine trial for their inquiry. They found that multi-view canonical correlation analysis (MCCA) and Group Principal Component Analysis (PCA) were equally superior in their ability to identify pertinent factors. No real insights into factors influencing vaccination response appear to have resulted from this study. The utility of the study seems to reside in its being an early attempt to apply machine learning approaches to vaccinology questions that may be useful to refining ML methods in the field.

Point 1: The authors should add a table of abbreviations. 

Response 1: We wish to thank the reviewer for the helpful comments. Reviewer 3 cited the need to reduce the number of times in which a keyword is defined as well. We could not find indicated in the guidelines where to position the table. We have thus added a table of abbreviations after the conclusions in Appendix A.

Point 2: A better explanation of ML approaches to biomedical problems would also be helpful.  

Response 2: We hope to have successfully addressed this point by adding a paragraph in the introduction that serves as a brief explanation of the need and utility of ML and AI in the biomedical field. It reads:

In recent years the field of biomedical research has witnessed an exponential growth in data availability, complexity and heterogeneity stemming from advancements in high-throughput technologies. To reap the benefits hidden within this data, traditional analysis methods alone do not suffice. Machine learning has emerged as a powerful tool for biomedical researchers to automatically learn from and make predictions based on data. By exploiting this technology, researchers can handle multidimensional datasets with data-driven strategies tailored to biomedical problems. Through artificial intelligence, the inherent heterogeneity and noise often encountered can be tackled to extract valuable insights otherwise difficult to discern, holding great promise for clinical applications.

Point 3: Although it may have been addressed in the original reports of this trial, it would be helpful to describe how HBs titers = 100 relate to HBV protection.  

Response 3: We are unsure if Reviewer 2 intended to indicate an antibody titer of 10 or 100 in their review. We will try to address both to avoid falling into error.

Anti-HBs titers of 10IU/L have been identified as a threshold for seroconversion and are commonly used to determine protection in a variety of hepatitis B studies (e.g. Hadler et al., 1986 doi: 10.1056/NEJM198607243150401)

We have also clarified the use of this threshold directly in the text in the Study cohort subsection of the Materials and Methods section. The modified paragraph reads:

The vaccine response was measured based on antibodies against HBV surface antigen (anti-HBs) titres captured by ELISA assays at days 0, 60, 180 and 365 (summarised in Table 1). To assess seroconversion, and thus presumed protection, a threshold of 10IU/L is commonly considered [19]. For each patient, if a level of anti-HBs of at least 10IU/L was reached before two months, early conversion was determined. If this value was induced not later than 6 months after the first dose, late conversion was registered for the individual. The patient was otherwise considered a non-converter.

Regarding anti-HBs titers of 100IU/L: in the first study conducted, an additional breakdown of seroconverters took in consideration the levels of anti-HBs to divide responders into high and low responders. An empirical limit of 100IU/L was chosen in that case. For the present study however, we focused mainly on the delay of the response rather than on the sheer quantity of antibodies induced.

Point 4: It is surprising that B cells are strongly negatively related to HBV protection (Fig. 9).  A discussion of this would be interesting.  Is it related to methods of assessing B-cell function?

Response 4: The reviewer makes a good point, that B cell gene expression is negatively related to antibody titers. However, in this study, we make a case for an activated immune system at baseline (before vaccination) to be responsible for a poor response to the vaccination challenge. We have added a paragraph in the discussion section to specify and emphasize this, it reads:

Furthermore, the negative association between seroconversion and B-cell-related gene expression may seem counter-intuitive, given their role as antibodies producers. However, this finding can be reconducted to the overall consideration that an activated immune system at baseline is, at least partially, responsible for a poor response to vaccination.

Reviewer 3 Report

The authors present a retrospective analysis of a HBV vaccine trial to test the ability of different multivariate statistical analyses to identify correlates for responders and non-responders. The paper is in general well-reasoned and described and requires only minor edits prior to publication. The authors have established via this analysis that multivariate techniques can extract trends from the analyzed dataset that comport with known markers for response and non-response to HBV vaccination. Moreover, they identified single view PCA as the one technique that failed to meet their performance criteria, which is useful to establish going forward with additional datasets, in particular those designed with this type of analysis in mind. Their methods can then be used to examine both B and T cell responses along with other predictive data as described in this manuscript. 

The following areas need to be addressed prior to publication:

1. The figures in general need to be larger so they are easier to read.

2. The authors state the observed correlates upfront in the abstract but not in the final paragraph of the introduction. They should restate them in the intro.

3. Some terms like AUC, PCA, etc. are defined multiple times. They should only be defined once.

The English is fine and requires only standard copy editing one would expect for any manuscript.

Author Response

The authors present a retrospective analysis of a HBV vaccine trial to test the ability of different multivariate statistical analyses to identify correlates for responders and non-responders. The paper is in general well-reasoned and described and requires only minor edits prior to publication. The authors have established via this analysis that multivariate techniques can extract trends from the analyzed dataset that comport with known markers for response and non-response to HBV vaccination. Moreover, they identified single view PCA as the one technique that failed to meet their performance criteria, which is useful to establish going forward with additional datasets, in particular those designed with this type of analysis in mind. Their methods can then be used to examine both B and T cell responses along with other predictive data as described in this manuscript.

The following areas need to be addressed prior to publication:

Point 1: The figures in general need to be larger so they are easier to read.

Response 2: We wish to thank the reviewer for their analysis and their comments. The figures have been made larger to comply with the suggestion.

Point 2: The authors state the observed correlates upfront in the abstract but not in the final paragraph of the introduction. They should restate them in the intro.

Response 2: We agree with the comment and we have restated the seroconversion correlates in a small paragraph added at the end of the introduction, it reads:

The markers, deriving from different data types, include clinical factors such as age and gender as well as inflammatory gene sets and vaccine specific T-cells predating vaccination challenge.

Point 3: Some terms like AUC, PCA, etc. are defined multiple times. They should only be defined once.

Response 3: We have addressed this point by removing duplicate definitions, in particular from figures’ captions, and by compounding this suggestion with Reviewer 1’s advice. We have thus added a table of abbreviations, after the conclusions, to clarify and minimize redundancies. Acronyms are still being introduced separately in the abstract, figures and text when first encountered.
Small copy-editing changes have also been performed.

Reviewer 4 Report

The article is quite interesting, with its main objective being to evaluate the capacity of multivision modeling on the responsiveness to vaccination against Hepatitis B virus (HBV). Although this objective is addressed in the text, there are some points that require attention. Below are some of these points:

-The number of samples seems small for some of the statements made throughout the text. It is also not clear why four samples are absent in some of the analyses conducted in the manuscript.

-It would be important to include the age of the groups represented in Figure 1.

-The statement "Age is undoubtedly an important factor in immune response, especially with 'inflammaging' and 'immunosenescence' being well-documented phenomena" seems difficult to relate to the presented data, as there is not a wide age distribution. It would be better to indicate that the data shows a tendency towards this assertion.

-The article mentions the paradox of Simpson, but further information is needed to better explain its relation to the provided information.

- I believe the conclusion should be more definitive regarding the effectiveness of multivisualization modeling, highlighting the positive aspects found.

Overall, the article has great potential but requires adjustments in methodology, improvement in the discussion section (especially regarding the encountered difficulties), and clearer emphasis on the vaccination response in the conclusion.

The quality of English in the provided text is good. The sentences are well-structured, and the vocabulary is appropriate for the scientific context. Overall, the text is well-written and effectively conveys the purpose and findings of the study.

Author Response

The article is quite interesting, with its main objective being to evaluate the capacity of multivision modeling on the responsiveness to vaccination against Hepatitis B virus (HBV). Although this objective is addressed in the text, there are some points that require attention. Below are some of these points:

Point 1: The number of samples seems small for some of the statements made throughout the text. It is also not clear why four samples are absent in some of the analyses conducted in the manuscript.

Response 1: We wish to thank the reviewer for their suggestions. The reviewer puts forth a valid criticism, however, to the best of our knowledge, the number of samples available to us is a commonly used number of samples in this type of vaccination study. We acknowledge the need for further validation of the methodology in the conclusions with the following statement: “To confirm the utility of this modelling, we recommend testing the methodology on other HBV vaccination cohorts or in other immunological settings”.  Furthermore, we chose to implement a leave-one-out cross validation for performance estimation to make optimal use of the data available.

We chose to discard the non-converters for two main reasons. Firstly TCR sequencing data was not available for one of the samples, further limiting the number of samples belonging to this class to 3. Secondly we used SMOTE to upsample the minority class to have a balanced classification problem. However, with such a low amount of data, the majority of the samples used for training would have been generated synthetically, making the results less trustworthy.
This has been clarified in the text, which now reads:

For the non-converter class, it is impossible to determine if seroconversion was eventually reached. Moreover, TCR data was available for only three non-converters. Therefore, this class was even more limited in size, further hindering the use of upsampling techniques to balance the task. Non-converters were thus excluded from further analysis.

Point 2: It would be important to include the age of the groups represented in Figure 1.

Response 2: The age ranges have been clarified and added to the caption of Figure 1 as follows:

Average (range) age, in years, per seroconversion class: early-converters, 35.9 (21.3 – 50.2); late-converters, 44.19 (36.3 – 48.5); non-converters, 23.53 (21.6 – 27.2).          

Point 3: The statement "Age is undoubtedly an important factor in immune response, especially with 'inflammaging' and 'immunosenescence' being well-documented phenomena" seems difficult to relate to the presented data, as there is not a wide age distribution. It would be better to indicate that the data shows a tendency towards this assertion.

Response 3: The reviewer makes a good point. Following the remark, this has been clarified with a statement in the discussion. It reads:

Even though the age distributions of early and late-converters are not substantially dissimilar, the data indicates a tendency that supports the argument of a higher risk of malfunctioning of the immune system in the elderly.

The paragraph has been consolidated and reinforced with the response to one of Reviewer’s 2 comments.

Point 4: The article mentions the paradox of Simpson, but further information is needed to better explain its relation to the provided information.

Response 4: This has been clarified in the text, which now reads:

This known effect, caused by multicollinearity in regression analysis, produces a change in size or sign of the contributions of two positively correlated variables when considered in association with other variables. The effect warns against the easy interpretation of the results and suggests the need for a causal analysis.

Point 5: I believe the conclusion should be more definitive regarding the effectiveness of multivisualization modeling, highlighting the positive aspects found.

Response 5: The conclusion has been reworked to highlight the strengths of the findings and of the methodology implemented. It now reads:

Our multi-view modelling reflected many aspects of what is known about natural HBV infections when giving a holistic and systemic perspective of HBV vaccination response data. Thereby, this methodology could recapitulate the known findings present in the literature, mirroring HBV response signatures and uncover novel markers. The results presented throughout this paper suggest the methodology is valid to identify biologically important features and why multi-level interrogations can aid in finding the most critical variables. This study’s approach identified that early converters have more in common with those that respond well to the natural infection, while late responders were associated with older individuals and an inflammatory phenotype. In particular, an overactivation at baseline of the innate component was detrimental to seroconversion while an early existing vaccine specific population in the T cell compartment fostered an earlier response. To confirm the utility of this modelling, we recommend testing the methodology on other HBV vaccination cohorts or in other immunological settings. We envisioned this type of framework will become useful for guiding rational vaccine antigen discovery as well as for the use of targeted adjuvants, and we hope it can pave the way for an advancement of immunological understanding.

Overall, the article has great potential but requires adjustments in methodology, improvement in the discussion section (especially regarding the encountered difficulties), and clearer emphasis on the vaccination response in the conclusion.

We hope to have effectively complied with the suggested changes by taking into consideration the reviewers’ comments, point by point.

Round 2

Reviewer 1 Report

The manuscript has been sufficiently improved by the authors to be published.

Reviewer 4 Report

The authors complied with the requested suggestions, making the text better presented, thus being plausible for publication. I only ask the authors for minor textual corrections, but nothing that changes the understanding of the work.

It just needs minor adjustments, mainly about the tense in some paragraphs. But nothing that changes the understanding of the work